# The Inhibitory Effect of *Pseudomonas stutzeri* YM6 on *Aspergillus flavus* Growth and Aflatoxins Production by the Production of Volatile Dimethyl Trisulfide

**DOI:** 10.3390/toxins14110788

**Published:** 2022-11-11

**Authors:** An-Dong Gong, Yin-Yu Lei, Wei-Jie He, Yu-Cai Liao, Ling Ma, Tian-Tian Zhang, Jing-Bo Zhang

**Affiliations:** 1Molecular Biotechnology Laboratory of Triticeae Crops, Huazhong Agricultural University, Wuhan 430070, China; 2College of Life Science, Xinyang Normal University, Xinyang 464000, China; 3College of Plant Science and Technology, Huazhong Agricultural University, Wuhan 430070, China; 4Hubei Key Laboratory of Plant Pathology, Wuhan 430070, China

**Keywords:** *Aspergillus flavus*, aflatoxins, *Pseudomonas stutzeri*, dimethyl trisulfide, post-harvest control

## Abstract

*Aspergillus flavus* and the produced aflatoxins cause great hazards to food security and human health across all countries. The control of *A. flavus* and aflatoxins in grains during storage is of great significance to humans. In the current study, bacteria strain YM6 isolated from sea sediment was demonstrated effective in controlling *A. flavus* by the production of anti-fungal volatiles. According to morphological characteristics and phylogenetic analysis, strain YM6 was identified as *Pseudomonas stutzeri.* YM6 can produce abundant volatile compounds which could inhibit mycelial growth and conidial germination of *A. flavus*. Moreover, it greatly prevented fungal infection and aflatoxin production on maize and peanuts during storage. The inhibition rate was 100%. Scanning electron microscopy further supported that the volatiles could destroy the cell structure of *A. flavus* and prevent conidia germination on the grain surface. Gas chromatography/mass spectrometry revealed that dimethyl trisulfide (DMTS) with a relative abundance of 13% is the most abundant fraction in the volatiles from strain YM6. The minimal inhibitory concentration of DMTS to *A. flavus* conidia is 200 µL/L (compound volume/airspace volume). Thus, we concluded that *Pseudomonas stutzeri* YM6 and the produced DMTS showed great inhibition to *A. flavus*, which could be considered as effective biocontrol agents in further application.

## 1. Introduction

*A. flavu*s, as a harmful phytopathogen, can infect peanuts, maize, and soybeans as well as their products, and produce highly toxic aflatoxins in the field and storage conditions [1]. Aflatoxins are considered unavoidable contaminants of human food and animal feed by the US Food and Drug Administration (FDA), which contaminate over 25% of the world’s crops annually [1] and cause great hazards to food security and human health [2]. Aflatoxin B1 (AFB1) is the most toxic member of the mycotoxin group, and its toxicity is 10 times that of potassium cyanide and 68 times that of arsenic [3]. The AFB1 is classified as a group I carcinogen by the International Agency for Research on Cancer (IARC) for its high degree of toxicity, and it may cause liver and lung carcinogen, and even acute death in humans [4].

It is estimated that more than 5 billion people are exposed to the harm of aflatoxins annually [5]. In the 1960s, hundreds of thousands of turkeys died from the aflatoxin-contaminated grain diet, which was known as the proverbial “Turkey X disease” [6]. Aflatoxins also caused serious economic losses and health hazards. For example, in the USA, financial losses due to aflatoxin contamination were estimated to be hundreds of million dollars annually, with maize and peanut being the most seriously affected food crops [7]. Of the 550,000–600,000 de novo liver cancer cases worldwide each year, it is estimated that 25,200–155,000 may be associated with aflatoxin exposure [8]. Countries considered endemic for hepatitis B virus infection (such as China, India, etc.) experienced more hepatocellular carcinoma and high death rates caused by aflatoxin [9]. For example, China has one of the highest populations of HBV carriers in the world, i.e., nearly 10% of the whole population. It was reported that over 370,000 people die of liver cancer every year, which accounts for more than 50% of the world’s liver cancer deaths [9].

Because of the great risks of aflatoxins to humans, more than 100 countries have legislatively defined the maximum levels in grains. The maximum tolerable limits for aflatoxins in maize and groundnut in the EU, the East African Community (EAC), and the USA are 4, 10, and 20 μg/kg, respectively [10,11]. In China, the maximum level for AFB1 is 20 ppb, with no regulations for total aflatoxins until now [5].

Legislations aimed to reduce the potential damage of aflatoxins are necessary. However, the more effective methods to avoid risks are preventing *A. flavus* growth and aflatoxins production in the field and storage. The methods used in controlling *A. flavus* and aflatoxins were the breeding of resistant plant varieties, chemical fungicides, and biological control agents. However, owing to the limited resistant breeds in nature and the lengthy breeding process, the effort to control the disease through plant breeding is less effective. While the approach of chemical fungicides is an efficient and quick controlling method against *A. flavus* and aflatoxin compared to plant breeding, it can cause pathogen resistance and pose potential health risks with pesticide residue [12]. It is worth noting that biological agents were considered as candidate alternatives in controlling *A. flavus* and aflatoxins in the field and storage. Some research has shown that certain microbes, such as non-aflatoxigenic *A. flavus* strain and bacteria, can prevent *A. flavus* infection and aflatoxins production in the field [13,14]. These microbes have been gradually utilized in controlling *A. flavu*s and aflatoxins in practice. Additionally, the microbial metabolites produced by microorganisms such as peptides, iturins, fengycin, and surfactants were also effective against *A. flavus* [15,16]. Among the identified metabolites, low-molecular-weight volatile compounds can quickly be evaporated and evenly distributed throughout the grain storage space [17]. In our previous work, the microbes including *Staphylococcus saprophyticus* L-38, *Serratia marcescens* Pt-3, *Enterobacter asburiae* Vt-7, and *Alcaligenes faecalis* N1-4 were demonstrated useful in controlling *A. flavus* and aflatoxins during storage by the production of antifungal volatiles [5,18,19,20]. In an effort to identify more novel strains for biocontrol, we isolated and identified one bacterial strain of *Pseudomonas stutzeri* YM6 from the collected marine sediment. The abundant anti-fungal volatile dimethyl trisulfide (DMTS) produced by YM6 showed strong anti-fungal activity against *A. flavus* and seven other important fungal pathogens in face-to-face dual cultural tests. It also greatly inhibited *A. flavus* growth and aflatoxins production in maize and peanut kernels during storage. The minimal inhibitory concentration of DMTS to *A. flavus* was 200 µL/L. The novel agents identified in the previous study can control *A. flavus* growth and aflatoxin production in grains during storage.

## 2. Results

### 2.1. Identification of Strain YM6

The colony of strain YM6 was opaque and pale yellow with a smooth surface and entire edge on the NA medium. YM6 was identified as a gram-negative, short-rod bacterium under the microscope. The 16S rDNA sequence of YM6 was homologous to three species (*P. stutzeri*, *P. putida*, and *P. xanthomarina*) of *Pseudomonas* spp. Eight strains of three species with great similarity to YM6 were selected to construct the phylogenetic tree. Strain YM6 and *P. stutzeri* 28a42 (AJ312165.1) showed the highest level of homology in terms of clade-level. Thus, the YM6 was identified as a strain of *P. stutzeri* (Figure 1). Its sequence was submitted to the NCBI database (accession number KF135442).

### 2.2. Inhibition of Pseudomonas stutzeri YM6 against A. flavus

Strain YM6 can greatly inhibit the growth of *A. flavus* in face-to-face dual culture without contact. The growth diameter of *A. flavus* inoculated on PDA medium was 4.5 cm at 5 dpi (days post inoculation). In the treatment group, YM6 inoculated on the NA medium produced abundant volatiles and greatly inhibited the mycelia growth of *A. flavus* in the PDA medium. The mycelia showed no signs of growth with YM6 treatment (Figure 2A). The inhibitory rate of volatiles from YM6 on mycelia growth of *A. flavus* was 100% at 5 dpi.

Similar results were also observed in the inhibitory tests of YM6 on conidia germination of *A. flavus*. In the control group, the *A. flavus* conidia inoculated on PDA medium germinated quickly, producing germ tubes that grow to form hyphae in 12 h (Figure 2B). In the subsequent 12 h, the hyphae branched outward, forming an extensive network that resembles the branches of a tree, and covered the surface of the PDA medium (Figure 2B). In the treatment group, the volatiles from YM6 inhibited the germination of *A. flavus* conidia with no formation of germ tubes during 24 h (Figure 2B). Thus, the inhibitory rate of volatiles from YM6 on conidia germination of *A. flavus* was 100% in dual culture.

### 2.3. Biocontrol Activity of YM6 against A. flavus and Aflatoxins in Peanut and Maize during Storage

In the control groups, the peanut and maize kernels were severely infected by *A. flavus*. There were a large number of green mycelia of *A. flavus* on the surface of peanut and maize seeds. Additionally, the number of infected kernels at a higher level of water activity (a_w_) outnumbered that at a lower level of a_w_. At a_w_ of 0.9, the disease incidence was 100% in the control groups of maize and peanuts. At a_w_ of 0.8 and 0.7, the disease incidence of the peanuts group was 82%, and of the maize control group was 32% (Figure 3A,B). In comparison, the infection of *A. flavus* was greatly inhibited in the presence of YM6. At higher a_w_ (0.9), peanut kernels were slightly infected by *A. flavus* with a disease incidence of 32%, while no infection was observed in the maize group. At lower a_w_ (0.7 and 0.8), no disease symptom was observed in maize and peanut kernels (Figure 3). The results clearly showed that the disease incidence rate in the control group was significantly higher than that in the YM6 treatment group. Based on these observations, we conclude that the volatiles from YM6 can greatly inhibit the infection of *A. flavus* conidia, and significantly prevent the disease development in peanut and maize kernels.

The aflatoxins in peanut and maize samples were also determined through quantitative analysis. In the control group, the total amounts of aflatoxins in peanut samples were 99.49, 330.17, and 1767.61 ppb at a_w_ of 0.740, 0.859, and 0.923, respectively. Similar phenomenon was also observed in maize samples that the content of aflatoxins was 27.09, 178.39, and 466.13 ppb at a_w_ of 0.785, 0.866, and 0.934, respectively (Table 1). On the one hand, these results showed that *A. flavus* produced more aflatoxins (with maximum AFB1 of over 80%) in crop seeds at higher a_w_ in the control group. On the other hand, no aflatoxin was detected in maize samples under three a_w,_ and peanut samples under a_w_ 0.740, 0.859 with the treatment of YM6. A little amount of aflatoxin (3.74 ppb) was detected in peanuts at a_w_ of 0.923 (Table 1). Therefore, the volatiles from YM6 are able to inhibit the production of aflatoxins by *A. flavus* in peanut and maize samples at higher a_w_ under storage conditions.

### 2.4. Analysis of A. flavus Cell Structure Affected by YM6

The experiment inoculated peanuts with conidia of *A. flavus* at a_w_ of 0.923 and cultured them at 28 °C for 5 days. The phenotype of conidia was analyzed through a scanning electron microscope and the results indicated conidial germination and transition from conidia to hyphae in the control group. The peanut seed coat was found to be covered by abundant mycelia. The hyphae also formed conidiophores and produced large amounts of conidia which induced secondary infection. The conidia were uniform in shape with fertile spherical proliferations on the surface. In contrast, only a few severely dehydrated conidia were found on the surface of peanuts in the YM6 group. The conidia were too deformed to germinate into hyphae (Figure 4). The observation suggested that under non-contact conditions, the volatiles from YM6 can effectively prevent *A. flavus* infection and destroy the cell structure of *A. flavus* conidia.

### 2.5. Chemical Identification of Volatiles Produced by the YM6 Strain

GC-MS analysis showed that the YM6 strain could produce abundant volatile substances (Figure 5). These volatiles were dimethyl trisulfide, 1-(trimethylsilyl)-1-propyne oxalic acid, 1-methyl-2-pentyl-cyclohexane, isobutyl pentadactyl ester, undecane, and 3,5-dimethyl-Isoxazole (Table 2). These substances with a molecular weight ranging from 97 to 356 Dalton (D) can easily volatilize. Only one type of substance was characterized based on comparison with the library NIST 08, with a similarity higher than 90% and large relative abundance (over 1%, peak area/sum area of all peaks), which was considered as a candidate volatile from YM6 and the key to the anti-fungal activity. The volatile was further identified to be DMTS based on the comparison of retention time and fragment ions with the commercial standard.

### 2.6. Minimal Inhibitory Concentration of DMTS against A. flavus

DMTS purchased from Sigma was used to test the inhibitory effect against *A. flavus*. The result revealed that DMTS could significantly inhibit the growth of *A. flavus* in confined spaces. The mycelia of *A. flavus* spread quickly to the edge of the Petri dishes and produced abundant green conidia at 5 dpi in the control group. Additionally, DMTS showed increased antagonistic effects against *A. flavus* with elevated concentrations. At a lower concentration of 50 µL/L (compound volume/airspace volume), the inhibition rate was only 7.5%. At 100 µL/L, the inhibition rate was 20.93%. DMTS completely inhibited the growth of *A. flavus* at 200 µL/L. Thus, the minimal inhibitory concentration (MIC) for DMTS against *A. flavus* was 200 µL/L (Figure 6).

## 3. Discussion

*A. flavus* is a globally distributed saprophytic fungus that infects many important crops, such as maize, peanuts, and soybeans during storage [20]. Additionally, it produces highly toxic and carcinogenic aflatoxins which threaten food security and human health. Hence, identifying safe and efficient agents to control *A. flavus* and aflatoxins in crops during storage is of significance and worth further study.

Microbes as important organisms are widely distributed in the natural environment. They can produce abundant secondary metabolites which have been *extensively* used in control plant pathogens such as *Fusarium graminearum* [21], *Penicillium digitatum* [22], *Rhizoctonia solani* [23], *Penicillium digitatum* [24], and *A. flavus* [25]. It is difficult to prevent *A. flavus* infection and aflatoxins production in grains during storage as *A. flavus* possesses great spore production, discharge, and dispersal capacities. Many chemical fungicides, although effective in controlling *A. flavus* and aflatoxins in storage, exhibit detrimental effects on food security and human health due to food contamination caused by pesticide residue. Compared to stable pesticides, compounds with smaller molecular sizes evaporate faster and can be evenly distributed in the airspace of storage, which is crucial to the control of *A. flavus* and aflatoxins. In 2015, our lab first demonstrated that volatile organic compounds (VOCs) produced by *Shewanella algae* strain YM8 [26] could greatly inhibit *A. flavus* growth and aflatoxin production in grains during storage. Some other microbes also have been proved with the capacity to prevent *A. flavus* infection and aflatoxin production in grains such as *Streptomyces alboflavus* [27], *Streptomyces yanglinensis* [28], *Bacillus megaterium* [29], *Pseudomonas protegens* [29], *Staphylococcus saprophyticus* [18], and *Serratia marcescens* [19]. However, there was no report of volatiles from *Pseudomonas stutzeri* regarding their effects against *A. flavus* and aflatoxins.

In the current study, YM6, which is a strain of *Pseudomonas stutzeri,* was isolated from marine sediment. It can produce volatiles that can greatly inhibit *A. flavus* growth and aflatoxins production in storage. The volatiles from strain YM6 exhibited a significant anti-fungal effect as they could inhibit the mycelial growth and spore germination of *A. flavus* by 100%, and severely damage the cell structure of *A. flavus*. The key antimicrobial substance produced by YM6 was identified as DMTS, with a minimal inhibitory concentration against *A. flavus* being 200 µL/L.

We also found that DMTS could deform the conidia of *A. flavus* and prevent the cell germination, but the inhibitory mechanism of DMTS for *A. flavus* remains unknown. However, on the basis of the study by Tang et al. [30], DMTS could cause deterioration of subcellular structures of *Colletotrichum gloeosporioides*, such as cell walls, plasma membranes, Golgi bodies, and mitochondria, as well as contribute to the leakage of protoplasm and cell death. Moreover, Zuo et al. [31] found that, in *Fusarium oxysporum* cells, DMTS could affect the glycosylation and ROS accumulation, inhibit steroid biosynthesis and glycerophospholipids metabolism, disrupt the cell membrane integrity, and finally result in the cell death. Taken together, DMTS could enter into fungal cells, disrupt cell structures, and inhibit cell growth. These results provided important evidence for understanding the inhibitory mechanism of DMTS for *A. flavus*, whereas the molecular target as well as interacted proteins of DMTS have not been investigated in fungal cells, which is worth further investigation.

Many volatiles produced by microbes have been demonstrated effective in controlling *A. flavus* and aflatoxins during storage, such as DMTS, dimethyl disulfide, methyl isovalerate, 1-Pentanol, Phenylethyl Alcohol and 3, 3-Dimethyl-1,2-epoxybutane, et al. [5,18,20,26]. Among these, sulfide exhibits have been demonstrated as excellent biocontrol effects in controlling soil-borne diseases by effectively inhibiting several pathogens and nematodes in soil [32,33]. DMTS, as an active substance, is a colorless or pale yellow substance with a strong mint odor, which exists in fresh onions, Chinese chive [34], and produced by some microbes [26]. However, this paper is the first to report that DMTS was produced by *Pseudomonas stutzeri*. DMTS is commonly used as permissible food additives in seasoning, gravy, and soup in many countries. DMTS can be applied in the storage of grain and oil products for its fast evaporation and dispersal capacities, and even distribution in confinement spaces to control plant pathogens and mycotoxins.

In conclusion, we were the first to prove that strain YM6 of *Pseudomonas stutzeri* showed great inhibition on *A. flavus* and aflatoxins in grains during storage by producing volatile DMTS. As a result, we deduced that the bacterial strain of YM6 and one of its VOCs (DMTS) could be used predominantly as a bio-control agent for crop protection in post-harvest stage. In addition, the microbe-associated volatiles showed great antifungal activity to *A. flavus* in laboratory investigation, but it is still a long distance from lab to commercial application in grains during storage. Some important questions should be answered before commercial application. For example, different to the ventilated requirement in large-scale granaries and small barns nowadays, the application of volatiles developed here needed to be kept in airtight storage condition. Apart from that, some other queries should also be solved before commercial application such as the mass production and enrichment of volatiles, usage cost limitation, usage dosage and persistence, volatile residues, as well as biosecurity to livestock and humans. These limits presented here provide us with further research directions.

## 4. Materials and Methods

### 4.1. Microbes and Plants

Bacteria YM6 was isolated from sea sediment in the Yellow Sea of China. YM6 was cultured in nutrient agar (NA) medium at 37 °C in the dark for 24 h. *Aspergillus flavus* strain 535 isolated from diseased peanuts was stored in our lab and inoculated onto potato dextrose agar (PDA) medium and cultured at 28 °C in the dark for anti-fungal tests [26].

Peanuts (cultivar Silihong) were purchased from supermarkets and used for inoculation tests. Dimethyl trisulfide (DMTS, CAS: 03658-80-8) was purchased from Sigma (Sigma-Aldrich, St. Louis, MO, USA) and used as the chemical standard for volatile identification.

The experiments were carried out in Molecular Biotechnology Laboratory of Triticeae Crops, Huazhong Agricultural University, Wuhan, China.

### 4.2. DNA Extraction and Phylogenic Analysis

The strain YM6 was inoculated onto NB medium and cultured in a flask at 28 °C at 200 rpm in the dark for 24 h. The suspension was centrifuged at 12,000 rpm for 10 min. The collected cells were used for genomic DNA extraction [26]. Then, 16S rDNA was amplified by PCR and amplified fragments were sequenced by Shanghai Sangon Biological Technology Company with the following sequencing primers: 27F (AGAGTTTGATCCTGGCTCAG) and 1541R (AAGGAGGTGATCCAG CCGC). The PCR was performed under the following conditions: initial denaturation at 94 °C for 5 min; followed by 30 cycles of 94 °C for 30 s, 55 °C for 30 s, and 72 °C for 40 s; and then 72 °C for 10 min. The 16S rDNA sequences of YM6 were submitted and aligned in the GenBank database. The 16S rDNA of bacteria with similarity over 95% to the sequences of YM6 were used for the construction of a phylogenic tree. The phylogenic tree of homologous strains was constructed by MEGA software with the neighbor-joining method.

### 4.3. Inhibitory Effect of YM6 on Mycelial Growth of A. flavus

A single YM6 colony was streaked over the NA medium surface and cultured at 37 °C in the dark for 24 h. The sterile water was added in NA medium to wash off the bacteria cells on the NA surface with an adjusted concentration of 10^8^ CFU/mL for a further anti-fungal test.

*A. flavus* was cultured on PDA medium for 4 days and produced abundant conidia. The conidia were then inoculated in PDB medium and cultured at 28 °C at 200 rpm for 3 days to produce mycelial pellets. Fresh *A. flavus* conidia and mycelial pellets were used in the face-to-face dual cultural test to analyze the antagonistic activity of YM6.

The regular size of *A. flavus* pellets were inoculated in the center of PDA plates, respectively. Fresh YM6 cells (100 µL, 10^8^ CFU·mL^−1^) were spread on the surface of the NA plate. Then, the PDA plate inoculated with *A. flavus* pellet was placed above the NA plate containing YM6. The two plates were placed face-to-face and sealed with two-layer tapes. The PDA plate inoculated with *A. flavus* pellet and challenged with NA medium was used as a control. Each treatment was repeated three times. All plates were cultured at 28 °C in the dark for 5 days. The mycelium diameters of *A. flavus* and the inhibition rate of YM6 in these treatments were calculated as follows:Inhibition rate (%) = [(mycelium diameter in control group − mycelium diameter in YM6 group)/mycelium diameter in control group] × 100.

### 4.4. Inhibitory Effect of YM6 on the Conidia Germination of A. flavus

FTF dual cultural test was used to analyze the inhibitory effect of YM6 on conidia germination of *A. flavus*. A total of 100 µL of *A. flavus* spores (5 × 10^5^ CFU/mL) was evenly coated onto cellophane covering the surface of the PDA plate, and the YM6 (100 µL, 10^8^ CFU/mL) was coated onto the surface of the NA plate. The two plates were placed FTF, and the PDA plate was inoculated with *A. flavus* conidia on top. The PDA plates inoculated with *A. flavus* conidia challenged with NA medium were used as controls. Each pair of the plates was sealed with two-layer tapes and cultured at 28 °C in the dark for 24 h. The character characteristics of conidia on cellophane were detected on a microscope (Hitech Instruments Co. Ltd., Shanghai, China), and the germination rate of conidia in the control and YM6 group was determined. The inhibition rate was calculated as follows:Inhibition rate (%) = [(conidia germination rate in control − conidia germination rate in YM6 treatment)/control conidia germination rate] × 100.

### 4.5. Biocontrol Activity of YM6 against A. flavus and Aflatoxins in Peanut and Maize

Maize and peanut kernels (100 g each) were grouped into 3 sets respectively and transferred into six flasks (250 mL) to be sterilized at an environment of 121 °C and 1.01 MPa for 20 min. The kernels were then cooled to room temperature. Fresh *A. flavus* conidia (1 mL, 5 × 10^5^ CFU·mL^−1^) was added into each flask and mixed for 10 min. Three flasks containing peanut kernels were filled with sterilized water; the water activity was determined at 0.785, 0.866, and 0.934 through an electronic dewpoint water activity meter, Aqualab Series 3 model TE (Decagon Devices, Pullman, Washington, DC, USA), respectively [26]. The values of water activity in flasks containing maize were measured at 0.740, 0.859, and 0.923, respectively. The maize and peanut kernels were used to test the biocontrol activity of YM6. The kernels in each flask were equally grouped into two sets, with one set placed in one compartment of the Petri dish (two in total). The other compartment containing NA medium was spread with YM6 strain (50 µL, 10^9^ CFU/mL). The kernels in one compartment challenged with NA medium (in the other compartment) were used as controls. All Petri dishes were sealed and cultured at 28 °C in darkness. The disease incidence of peanut and maize was calculated at 7 dpi.

Aflatoxins in maize and peanuts were extracted with acetonitrile/water and analyzed through ultra-performance liquid chromatography and mass spectrometry (UPLC-MS, Thermo Scientific, New York, NY, USA) [26].

### 4.6. Structural analysis of A. flavus Treated by Volatile from YM6

*A. flavus* spores were inoculated to the peanut surface with a water activity of 0.934 and challenged with YM6 through FTF dual culture for 5 days. The *A. flavus* inoculated on peanuts surface without the presence of YM6 was used as a control. The peanut seeds were fumigated with osmic acid (0.1%, *v/v*) for 1 h, and then placed at room temperature for 3 h [35]. A small piece of the peanut coat (0.5 cm × 0.5 cm) was removed with a dissecting knife, clamped with stubs, and coated with a layer of gold for examination of SEM (JSM-6390, Hitachi Corporation, Tokyo, Japan) to observe the ultra-structure of *A. flavus*.

### 4.7. Identification of Volatiles from the Strain YM6

The strain YM6 was inoculated onto the surface of the NA medium in a 100 mL flask. The flask containing NA medium without YM6 inoculation was used as a control. All flasks were sealed with double-layer plastic film and cultured at 28 °C in the dark for 48 h. All the experiments were repeated twice. Then, the flasks were incubated in a water bath of 40 °C for 30 min. The volatile substance was extracted by solid phase micro-extraction (SPME, divinylbenzene/carboxy/polydimethylsiloxane) and analyzed by gas chromatography-mass spectrometry (GC-MS) (Agilent Technologies, Santa Clara, CA, USA). For volatile enrichment, the metal head of SPME was inserted through the plastic film into the flask. The coated fiber in the metal head was pushed out and placed in the center position above the NA medium to absorb the volatiles for 50 min. The coated fiber was retracted and transferred into the GC-MS system for volatile analysis. The detection parameter was used as below [26].

The splitless injection of GC was used in the GC-MS analysis. The carrier gas was helium; the inlet temperature was 250 °C. The oven was set for procedures as follows: started at 40 °C for 3 min; heated up to 160 °C at the rate of 3 °C/min for 2 min; and then raised to 220 °C at the rate of 8 °C/min for 3 min. The MS analysis was performed with an EI source under a temperature of 230 °C. The quadrupole temperature was set at 150 °C. The collision energy was 70 eV. The mass range was set from 50 to 500 amu. The detected volatiles were verified by aligning in the National Institution of Standards and Technology (NIST 08) database, and the retention time was compared with authentic compound purchased from the company.

### 4.8. Inhibition Effect of DMTS on A. flavus

The volatile substance produced by YM6 showed a significant antagonistic effect on *A. flavus* growth. DMTS (liquid state, purity ≥ 98%), the major component emitted by YM6, was purchased from Sigma and tested as the anti-fungal agent against *A. flavus*. The anti-fungal analysis was conducted through the FTF dual cultural test. A piece of circular filter paper (5 cm diameter), moistened by DTMS, was placed in a Petri dish. The final concentration of DMTS in each treatment was adjusted to 200 µL/L, 100 µL/L, and 50 µL/L, respectively. The other Petri dish containing PDA medium was inoculated with *A. flavus* mycelia. The dish containing *A. flavus* was placed FTF above the dish containing DMTS and sealed with two layers of tape. The dish of *A. flavus* co-cultured with the same volume of water was used as a control. All plates were cultured at 28 °C in darkness for 5 days. The mycelium diameter of *A. flavus* was measured and the inhibition rate was calculated.

### 4.9. Data Analysis

All experiments were conducted in duplicate. The amount of aflatoxin was evaluated and shown as mean ± SE. The significant differences between mean values were determined using Duncan’s multiple range test (*p* < 0.05) following one-way analysis of variance (ANOVA). The statistical analysis was performed using SPSS 16.0 software (SPSS Inc., Chicago, IL, USA).

## Figures and Tables

**Figure 1 toxins-14-00788-f001:**
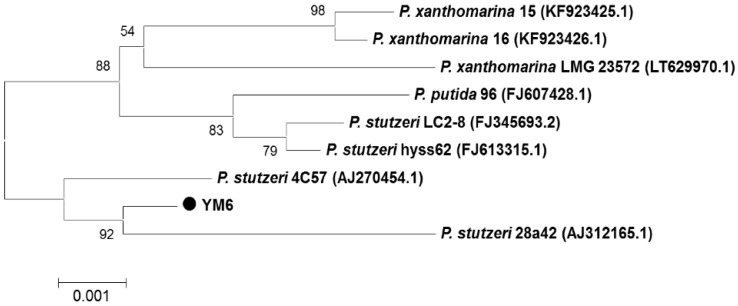
Neighbor-joining phylogenetic tree based on 16S rDNA sequence of strain YM6 and other homologous strains retrieved from the NCBI database. The scale bar indicates the number of substitutions per base position.

**Figure 2 toxins-14-00788-f002:**
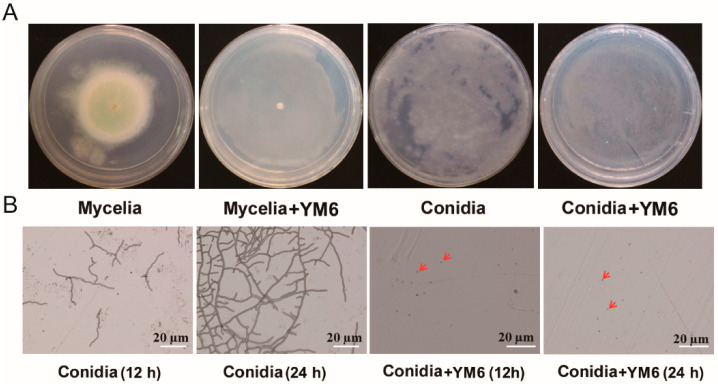
The inhibitory effect of strain YM6 on the growth of *A. flavus* conidia and mycelia in sealed Petri dishes. (**A**). The growth of *A. flavus* mycelia/conidia cultured on the PDA plate was completely inhibited by volatiles produced by YM6 on NA plate in face-to-face dual culture. The two dishes were placed face-to-face without physical contact and cultured at 28 °C for 5 days. (**B**). The germination of *A. flavus* conidia on PDA plate was completely inhibited by volatiles from YM6 under microscopy in 24 h. The red arrows point to the position of *A. flavus* conidia under microscopy.

**Figure 3 toxins-14-00788-f003:**
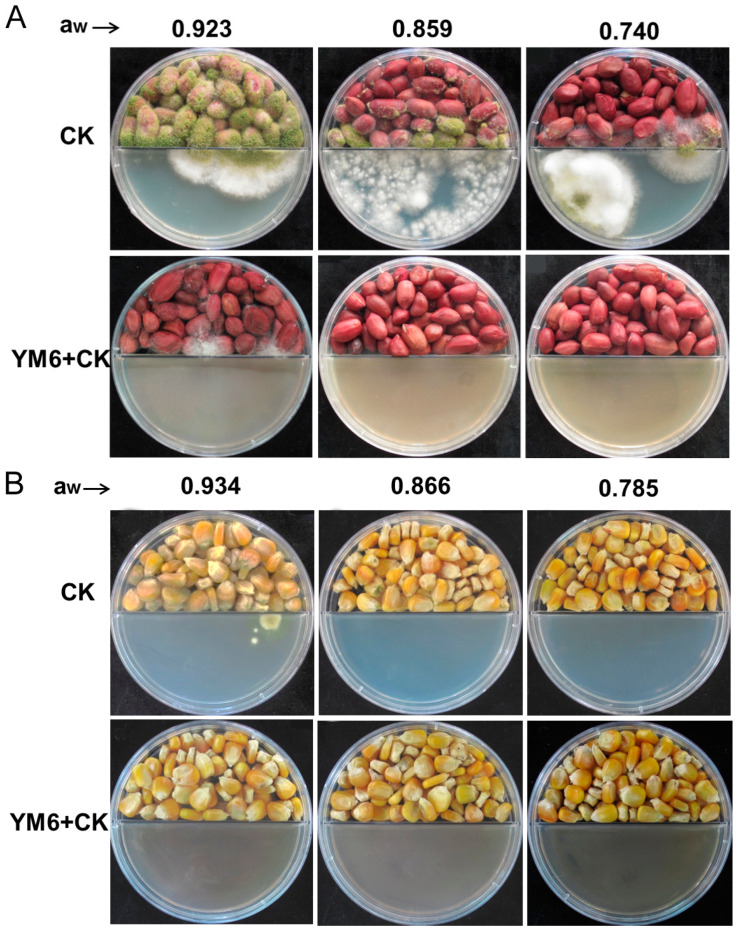
Biocontrol efficiency of strain YM6 against *A. flavus* infection in peanut (**A**) and maize (**B**) kernel. The grains of peanut and maize inoculated with A. flavus condia were placed on one side of separated Petri dishes, challenged with (*A. flavus* + YM6) or without (*A. flavus*) the presence of strain YM6 on the NA plate on the other side of the separated Petri dishes.

**Figure 4 toxins-14-00788-f004:**
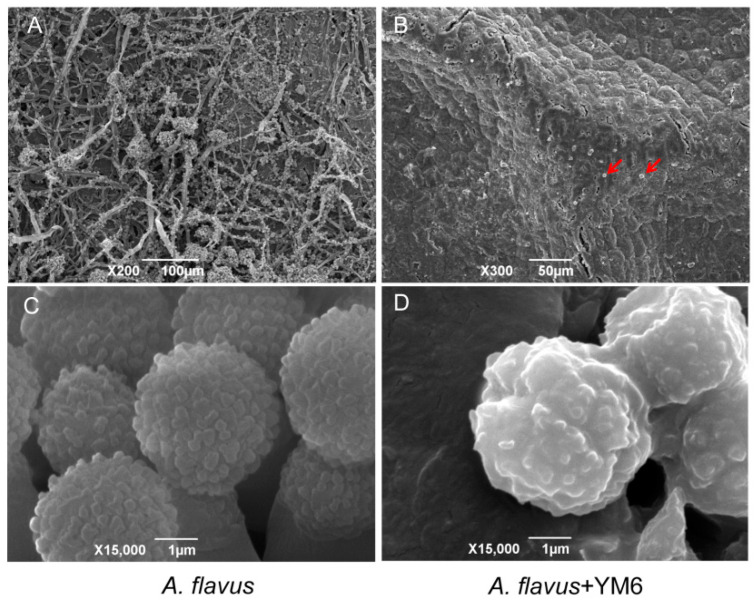
Ultra-structure of *A. flavus* conidia inoculated on peanut seeds affected by the volatiles from YM6. *A. flavus* conidia were inoculated on peanut seed coat and cultured for 5 days with (*A. flavus* + YM6) or without (*A. flavus*) the presence of YM6. In control group, the conidia on peanut coat could germinate to hyphae and phialids-conidiophores (**A**), and produce abundant conidia (**C**). The conidia on peanut surface cannot germinate to hyphae (**B**) and showed deformed character (**D**) in YM6 group. The red arrows pointed to the position of *A. flavus* conidia on the surface of the peanut coat.

**Figure 5 toxins-14-00788-f005:**
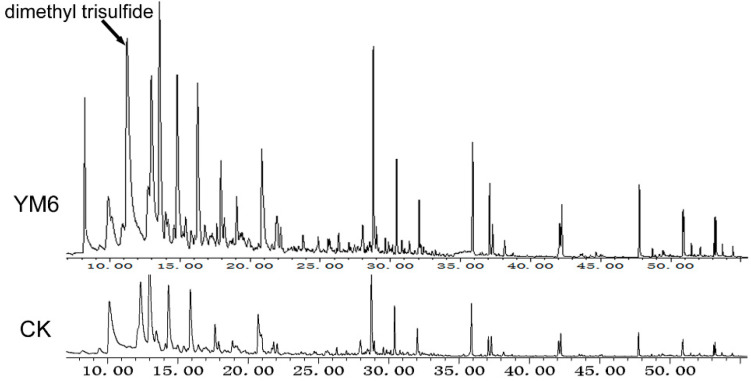
GC-MS analysis of volatiles emitted from strain YM6. Strain YM6 grown on NA plate for 48 h to produce volatile compounds. The compounds detected in YM6 spectra (YM6) omitting the same compounds in blank NA medium (CK) were considered the authentic compound.

**Figure 6 toxins-14-00788-f006:**
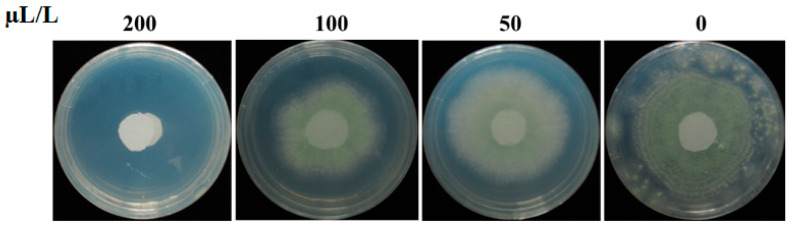
The minimal inhibitory concentration analysis of DMTS against *A. flavus* on PDA plates. The concentration of DMTS was 200, 100, and 50 µL/L (compound volume/airspace volume, *v*/*v*). The *A. flavus* conidia inoculated to the round paper in the center of the PDA plate was used as a control.

**Table 1 toxins-14-00788-t001:** Quantitative analysis of aflatoxins in inoculated maize and peanut kernels.

	Value of Water Activity	Treatment	Aflatoxin Concentration/ppb		
			AFB1	AFB2	Total
Peanut	0.740	Control	85.41 ± 3.21	14.08 ± 0.76	99.49
		YM6	—	—	—
	0.859	Control	299.36 ± 17.55	30.81 ± 7.83	330.17
		YM6	—	—	—
	0.923	Control	1575.19 ± 67.32	192.42 ± 9.22	1767.61
		YM6	3.74 ± 0.24 *	—	3.74
Maize	0.785	Control	24.92 ± 2.60	2.17 ± 0.13	27.09
		YM6	—	—	—
	0.866	Control	156.62 ± 4.63	21.77 ± 1.56	178.39
		YM6	—	—	—
	0.934	Control	402.31 ± 9.32	63.82 ± 4.35	466.13
		YM6	—	—	—

The concentration of aflatoxin was shown as average ± SE. Total means the sum of AFB1 and AFB2. — means that the aflatoxin was not detected with the minimum detection limit of 0.2 ppb. * means significant difference at *p* < 0.05 compared to control group.

**Table 2 toxins-14-00788-t002:** Identification of volatiles emitted from strain YM6 through GC-MS system.

No.	Compounds	Retention Time/min	Peak Area /% ^a^	Similarity /% ^b^	Mass /m·z^−1^
1	Dimethyl trisulfide	11.238	13.877	90	125.963
2	1-(trimethylsilyl)-1-propyne	14.797	7.451	52	112.071
3	Oxalic acid, isobutyl pentadecyl ester	16.777	0.962	72	356.293
4	1-methyl-2-pentyl- Cyclohexane	17.131	0.122	76	168.188
5	Undecane	18.785	0.200	93	156.188
6	3,5-dimethyl- Isoxazole	30.447	0.884	72	97.053
7	2,4,4,6,6,8,8-Heptamethyl-2-nonene	30.824	0.485	60	224.252
8	Butylated hydroxytoluene	37.084	0.992	97	220.183

^a^ The peak area of the identified compound to the total area of all peaks. ^b^ the mass spectrum of identified compound compared to the spectrum of standard substance in NIST 08 database, respectively. Mass means mass to charge ratio.

## Data Availability

Not applicable.

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
