# Peer review of "The Inhibitory Effect of *Pseudomonas stutzeri* YM6 on *Aspergillus flavus* Growth and Aflatoxins Production by the Production of Volatile Dimethyl Trisulfide"

_toxins, 2022, doi:10.3390/toxins14110788_

Round 1

Reviewer 1 Report

toxins-1997331-peer-review-v1

Reviewer summary: The goal of this work was to investigate and characterize the effect of dimethyl trisulfide (DMTS), a volatile compound produced by Pseudomonas stutzeri YM6 and its impact on Aspergillus flavus growth and aflatoxin production on maize and peanut kernels. Major findings were that (1) YM6 associated volatile metabolites were able to inhibit mycelial growth and conidial germination of Aspergillus flavus, (2) VOCs produced by YM6 also prevented fungal infection and aflatoxin production on maize and peanut kernels (3) SEM images provide some insides into possible mechanisms of YM6 volatiles (4) GC/MS analysis identified DMTS as the most abundant fraction produced by YM6 (13%) (5) MIC of DMTS on A. flavus conidia is 200 ug/L. Thus, authors concluded that Pseudomonas stutzeri YM6 and associated production of DMTS can inhibit A. flavus growth and aflatoxin contamination suggesting possible use as a biocontrol agent for control of aflatoxin post-harvest. Outcomes from this work highlighted the effectiveness of bacteria produced VOCs and its possible uses to control aflatoxin contamination during grain storage.

I have highlighted some strengths in this study as well as some comments and suggestions.

Strengths.

-       Authors demonstrated clear goals and motivation to conduct study

-       Strength in approach, for example molecular identification of YM6, phylogenetic construction, strong inhibition observed by YM6 volatile metabolites, proof-of-concept in YM6 effectiveness on maize and peanut kernels, SEM analysis of fungal cell structure, GC/MS for identification of DMTS, and MIC of DMTS against A. flavus

-       Also appreciated the authors including a standard laboratory treatment condition (PDA) to compare with maize and peanut kernels

Abstract

L8 proved – would recommend that authors avoid using the word proved, we cannot prove anything however, we can ‘show’ or ‘demonstrate’ or ‘provide support or evidence’, please modify use throughout text

L9 morphological characteristics

L13 proved – further support or demonstrate that

L15 to total peak area?

Introduction

L31-33 …10 times that of potassium cyanide and 68 times that of arsenic – how was this calculated? Please provide reference

L35 may cause liver cancer? It is a known liver and lung carcinogen

L38-39 please include reference for Turkey X disease

L41-42 please include reference for 25 million dollars loss in Georgia USA for peanuts

L44 Africa is not a country, instead it is a continent

L45 hepatocellular carcinoma

Not sure if authors had previously hypothesized on outcomes of research questions – might be worthwhile including hypothesis for this study

L58 proved

L58 non-aflatoxigenic

L60 ‘relatively safe for humans’ – how do we know this? this is not known unless there are toxicological assessments for safety, please provide support for this statement

L65 In our previous work, some efficient microbe with valid anti-fungal… consider rephrasing this sentence – what is effective and how effective? Some microbes? Which microbes?

L66 proved

Results.

L101 please include abbreviation for dpi

L144 how do authors differentiate between infection and disease development on kernels?

L189-190 please consider rephrasing sentence

L248 proved

Figure 2 there is not description for panel B in figure legend, was this microscopy? How did authors process this – not in methods?

Figure 3 has no aflatoxin production results within the images (please change figure legend) 

Figure 4 it would help the reader when authors include labels A-D for each panel and provide description

-       Authors do not provide methods for SEM, how were samples prepared, where they dried? Gold sputtered?

Figure 5 – please include and explain abbreviation CK

Table 1 – why are AF values only presented as single values, authors say that all treatments were run in triplicates, please provide data +/- SEM for aflatoxin levels detected by HPLC, did authors extract AF from each triplicate plate? Please include statistical analysis for Table 1

-       What does ‘no aflatoxin detected’ mean? 0 ppb – what is the limit of detection for HPLC? 

Discussion.

L282 proved

Authors should include a discussion on technical feasibility of using microbe-associated volatiles on grains in storage, why are they not used commercially? What are the technical and industrial barriers?

What about toxicity in humans?

L298 and 301 proved

L 295-297 Authors should discuss some potential limitations of this study, observations of 100% inhibition rate was on PDA – difference between standard laboratory conditions, grains, and actual storage, how different might the environment be

Others-observation on SEM – any discussion on potential mechanisms for DMTS on sporulation and observed cell structure differences?

Materials and Methods.

L310-320 – why do authors list these fungal strains, they were not used in this study

L344 – bacteria cells instead of bodies

L352 – PDA plates inoculated with A. flavus pellets was placed above NA plate containing YM6; how did authors prevent pellet from falling over onto YM6; where plates allowed to grow for some time before assembling FTF assay or immediately after inoculation

L370 Inhibition (spelling)

L377-378 – how was water activity adjusted – please specify and cite reference used

L393 – why were peanuts fumigated with osmic acid, please include brief rationale or reference

L416 – what is NIST 08, is 08 the version of the database?

L424 – DMTS purchased from Sigma is a solid powder? What was it dissolve in? is it water soluble? What was vehicle/control for MIC testing? Please include

L425 – Petri dish

Other comments-

1.     why was nutrient agar (NA) selected?

2.     What Aspergillus flavus strain was used – please provide strain designation, reference strain?

Author Response

Dear reviewers:

    On behalf of my co-authors, we thank you very much for giving us an opportunity to revise our manuscript, we appreciate reviewers very much for their constructive comments and suggestions on our manuscript. The point-to-point response to the reviewer 1 as follows:

summary: The goal of this work was to investigate and characterize the effect of dimethyl trisulfide (DMTS), a volatile compound produced by Pseudomonas stutzeri YM6 and its impact on Aspergillus flavus growth and aflatoxin production on maize and peanut kernels. Major findings were that (1) YM6 associated volatile metabolites were able to inhibit mycelial growth and conidial germination of Aspergillus flavus, (2) VOCs produced by YM6 also prevented fungal infection and aflatoxin production on maize and peanut kernels (3) SEM images provide some insides into possible mechanisms of YM6 volatiles (4) GC/MS analysis identified DMTS as the most abundant fraction produced by YM6 (13%) (5) MIC of DMTS on A. flavus conidia is 200 ug/L. Thus, authors concluded that Pseudomonas stutzeri YM6 and associated production of DMTS can inhibit A. flavus growth and aflatoxin contamination suggesting possible use as a biocontrol agent for control of aflatoxin post-harvest. Outcomes from this work highlighted the effectiveness of bacteria produced VOCs and its possible uses to control aflatoxin contamination during grain storage.

I have highlighted some strengths in this study as well as some comments and suggestions.

 Strengths.

-       Authors demonstrated clear goals and motivation to conduct study

-       Strength in approach, for example molecular identification of YM6, phylogenetic construction, strong inhibition observed by YM6 volatile metabolites, proof-of-concept in YM6 effectiveness on maize and peanut kernels, SEM analysis of fungal cell structure, GC/MS for identification of DMTS, and MIC of DMTS against A. flavus

-       Also appreciated the authors including a standard laboratory treatment condition (PDA) to compare with maize and peanut kernels

Abstract

L8 proved – would recommend that authors avoid using the word proved, we cannot prove anything however, we can ‘show’ or ‘demonstrate’ or ‘provide support or evidence’, please modify use throughout text

 Response: Thank you for your suggestion. It was changed to ‘demonstrated’ in L8. We also modified them throughout the whole text.

L9 morphological characteristics

Response: Thank you for your suggestion. The sentence was changed to ‘According to morphological characteristics….’.

L13 proved – further support or demonstrate that

 Response: Thank you for your suggestion. It was changed to ‘further supported that…’

L15 to total peak area?

 Response: Thank you for your suggestion. Total peak area means the sum area of all peaks detected GC-MS test. These words ‘(to total peak area)’ were deleted in the ‘abstract’, and elucidated in ‘results’ section as ‘sum area of all peaks’.

Introduction

 L31-33 …10 times that of potassium cyanide and 68 times that of arsenic – how was this calculated? Please provide reference

 Response: Thank you for your suggestion. The reference was added in the manuscript.

L35 may cause liver cancer? It is a known liver and lung carcinogen

Response: Thank you for your suggestion. It was changed to ‘liver and lung carcinogen’

L38-39 please include reference for Turkey X disease

 Response: Thank you for your suggestion. The reference was added in the manuscript.

L41-42 please include reference for 25 million dollars loss in Georgia USA for peanuts

 Response: Thank you for your suggestions. The sentence was not specific. It was changed to ‘in the USA, financial losses due to aflatoxin contamination was estimated to be hundreds of million dollars annual, with maize and peanut being the most seriously affected food crops’. The reference was added in the manuscript.

L44 Africa is not a country, instead it is a continent

Response: Thank you for your suggestion. The word ‘Africa’ was deleted.

L45 hepatocellular carcinoma

Not sure if authors had previously hypothesized on outcomes of research questions – might be worthwhile including hypothesis for this study

Response: Thank you for your suggestions. The phrase was changed to ‘hepatocellular carcinoma’. These messages were obtained in the reference of Chen et al. (2013). The reference was added in the tests.

L58 proved

Response: Thank you for your suggestion. The word ‘proved’ was changed to ‘showed’.

L58 non-aflatoxigenic

Response: Thank you for your suggestion. The phrase was changed to ‘non-aflatoxigenic’.

L60 ‘relatively safe for humans’ – how do we know this? this is not known unless there are toxicological assessments for safety, please provide support for this statement

Response: Thank you for your suggestions. The sentence ‘relatively safe for humans’ is not specific, and the safety to humans is not assessed till now. Thus, we deleted ‘are relatively safe for humans and’ and modified the sentence to ‘These microbes are relatively safe for humans and have been gradually utilized in controlling A. flavus and aflatoxins in practice.’

L65 In our previous work, some efficient microbe with valid anti-fungal… consider rephrasing this sentence – what is effective and how effective? Some microbes? Which microbes?

Response: Thank you for your suggestion. We have rephrased this sentence. The sentence was changed to ‘In our previous work, the microbes including Staphylococcus saprophyticus L-38, Serratia marcescens Pt-3, Enterobacter asburiae Vt-7 and Alcaligenes faecalis N1-4 were demonstrated useful in controlling A. flavus and aflatoxins during storage by the production of antifungal volatiles.’

L66 proved

Response: Thank you for your suggestion. The word ‘proved’ was changed to ‘demonstrated’.

Results.

L101 please include abbreviation for dpi

Response: The abbreviation of ‘dpi’ was added in the test.

L144 how do authors differentiate between infection and disease development on kernels?

Response: Thank you for your comment. The mycelia and conidia observed on peanut/maize seeds were considered as ‘infection by A. flavus’. The disease development means the greater number of infection seeds or higher disease incidence in the treatment. 

L189-190 please consider rephrasing sentence

Response: Thank you for your suggestion. The sentence was changed to ‘On the other hand, no aflatoxin was detected in maize samples under three aw, and peanut samples under aw 0.740, 0.859 with the treatment of YM6.’

L248 proved

 Response: Thank you for your suggestion. The word ‘proved’ was changed to ‘revealed’.

Figure 2 there is not description for panel B in figure legend, was this microscopy? How did authors process this – not in methods?

Response: Thank you for your suggestion. The descriptions for panel A, panel B and methods (section 4.4) were modified.

Figure 3 has no aflatoxin production results within the images (please change figure legend) 

 Response: Thank you for your suggestion. The description in the figure 3 legend was modified. The phrase ‘aflatoxin production’ was deleted.

Figure 4 it would help the reader when authors include labels A-D for each panel and provide description

-       Authors do not provide methods for SEM, how were samples prepared, where they dried? Gold sputtered?

Response: Thank you for your suggestions. The labels (A-D) were added in Figure 4. The description for labels was added in the legends. The method for SEM analysis was provided in section 4.6 of ‘material and methods’.

Figure 5 – please include and explain abbreviation CK

Response: Thank you for your suggestions. The CK means the spectra of blank NA medium. We added the explanation for CK in the figure 5 legend.

Table 1 – why are AF values only presented as single values, authors say that all treatments were run in triplicates, please provide data +/- SEM for aflatoxin levels detected by HPLC, did authors extract AF from each triplicate plate? Please include statistical analysis for Table 1

-       What does ‘no aflatoxin detected’ mean? 0 ppb – what is the limit of detection for HPLC? 

Response: Thank you for your suggestions. The data was re-checked and provided as data±SE in table 1. The statistical analysis between control and YM6 group was added. ‘no aflatoxin detected’ was changed to ‘the aflatoxin was not detected with the minimum detection limit of 0.2 ppb’ and added in the table 1 legend.

Discussion.

L282 proved

Authors should include a discussion on technical feasibility of using microbe-associated volatiles on grains in storage, why are they not used commercially? What are the technical and industrial barriers?

What about toxicity in humans?

Response: Thank you for your suggestions. It was changed to ‘demonstrated’. The message related to the usage barriers of microbe-associated volatiles on grains during storage was added in the tests.

L298 and 301 proved

Response: Thank you for your suggestions. It was changed to ‘demonstrated’.

L295-297 Authors should discuss some potential limitations of this study, observations of 100% inhibition rate was on PDA – difference between standard laboratory conditions, grains, and actual storage, how different might the environment be

Others-observation on SEM – any discussion on potential mechanisms for DMTS on sporulation and observed cell structure differences?

Response: Thank you for your suggestions. The potential limitations of this study were added in the discussion section. The message related to the inhibitory mechanism of DMTS was also added in the discussion.

Materials and Methods.

L310-320 – why do authors list these fungal strains, they were not used in this study

Response: Thank you for your suggestions. The broad spectrum antifungal test of YM6 was conducted with these fungal strains, but not provided in the current manuscript. The sentence was changed to ‘Aspergillus flavus strain 535 isolated from diseased peanuts was stored in our lab, and inoculated onto potato dextrose agar (PDA) medium and cultured at 28°C in the dark for anti-fungal tests.’

L344 – bacteria cells instead of bodies

Response: Thank you for your suggestions. The ‘bacteria bodies’ was changed to ‘bacteria cells’.

L352 – PDA plates inoculated with A. flavus pellets was placed above NA plate containing YM6; how did authors prevent pellet from falling over onto YM6; where plates allowed to grow for some time before assembling FTF assay or immediately after inoculation

Response: Thank you for your comments. The A. flavus pellets were cultured in PDB medium. So the pellet was soaked in medium. It can adhere to the surface of PDA plate by the surface tension of water. Another reason is that the surface of PDA plate was also wet, thus the pellet can not fall in a long time. Thus, the FTF assay was conducted immediately after inoculation. The test was conducted for many times, we seldom see the falling of pellets.    

L370 Inhibition (spelling)

Response: Thank you for your comments. The wrong spelling for inhibition was changed.

L377-378 – how was water activity adjusted – please specify and cite reference used

Response: Thank you for your comments. Different volume of water was added into the peanuts and maize samples, shaked for 10 min to make them uniform. The water activity was determined at 28◦C using an electronic dewpoint water activity meter, Aqualab Series 3 model TE (Decagon Devices, Pullman, Washington, USA). The detail message and reference were added in the manuscript.

L393 – why were peanuts fumigated with osmic acid, please include brief rationale or reference

Response: Thank you for your comments. The samples fumigated with osmic acid can fixed the cell structure, then the appearance of hyphae and mycelia was not changed in vacuum of SEM. The reference was added in the tests.

L416 – what is NIST 08, is 08 the version of the database?

Response: Thank you for your comments. NIST is a database in GC-MS system for compound identification, and 08 means the version of database.

L424 – DMTS purchased from Sigma is a solid powder? What was it dissolve in? is it water soluble? What was vehicle/control for MIC testing? Please include

Response: Thank you for your comments. DMTS purchased form Sigma was in liquid state. We added that in the manuscript. The MIC test of DMTS was conducted by using different volumes of DMTS reagent in same volume. 

L425 – Petri dish

Response: Thank you for your comments. The spelling ‘petri dish’ was modified.

Other comments-

  1. why was nutrient agar (NA) selected?

Response: Thank you for your comments. NA medium was commonly used for bacteria culture in our lab. Thus, we cultured strain YM6 in NA medium for growth and volatile production.

  1. What Aspergillus flavus strain was used – please provide strain designation, reference strain?

Response: Thank you for your comments. The strain number was added in the tests. The sentence was changed to ‘Aspergillus flavus strain 535 isolated from diseased peanuts was stored in our lab, and inoculated onto potato dextrose agar (PDA) medium and cultured at 28°C in the dark for anti-fungal tests.’

Best wishes.

Reviewer 2 Report

Dear editor,

In the present MS the authors Identified the most abundant volatile compound of Pseudomonas statutzeri YM6 (previously identified) and evaluated the antifungal effect of its volatile compounds. The authors also simulated a biocontrol experiment.

However, it is not clear the novelty of work, authors have already identified that this strain is antifungal and they have already tested it against Aspergillus in corn and peanuts, so the question is what is the novelty? Moreover, authors have never tested the compound DMST in isolation, it has always been a complex of volatile compounds produced by YM6 except the minimal inhibitory concentration. MS title leads to a misunderstanding and to make matter worse, the objective is not clear. Also, it is worth to note that tables and results lack statistical analyze/information. For these reasons, I believe the work is not suitable for publishing in Toxins.

Minor comments:

1. Introduction

-Avoid use of reference that are more than 10 years old.

-Lack citation line 39, 46 and 51.

-Include a paragraphy with information about DMTS

-Table 2 lacks statistical parameters. Convert ppb in mass/mass. Include information such as the number of samples, for example.

Author Response

Dear reviewer:

   On behalf of my co-authors, we thank you very much for giving us an opportunity to revise our manuscript, we appreciate reviewers very much for their constructive comments and suggestions on our manuscript. The point-to-point response to the reviewer 2 as follows:

In the present MS the authors Identified the most abundant volatile compound of Pseudomonas statutzeri YM6 (previously identified) and evaluated the antifungal effect of its volatile compounds. The authors also simulated a biocontrol experiment.

However, it is not clear the novelty of work, authors have already identified that this strain is antifungal and they have already tested it against Aspergillus in corn and peanuts, so the question is what is the novelty? Moreover, authors have never tested the compound DMST in isolation, it has always been a complex of volatile compounds produced by YM6 except the minimal inhibitory concentration. MS title leads to a misunderstanding and to make matter worse, the objective is not clear. Also, it is worth to note that tables and results lack statistical analyze/information. For these reasons, I believe the work is not suitable for publishing in Toxins.

Response: Thank you for your comments. That was helpful to us. The novelty of work is that Pseudomonas stutzeri YM6 was first demonstrated effective in control A. flavus and aflatoxins in grains during storage by producing volatile DMTS. The previous title was not specifie, as your advice, we modified the title to ‘The inhibitory effect of Pseudomonas stutzeri YM6 on Aspergillus flavus growth and aflatoxins production by the production of volatile dimethyl trisulfide’. The statistical analysis and detail information was added in the table and legends.

-Avoid use of reference that are more than 10 years old.

Response: Thank you for your comments. Some references more than 10 years old was deldeted. Some new references were added.

-Lack citation line 39, 46 and 51.

Response: Thank you for your comments. The related references were added in the test.

-Include a paragraphy with information about DMTS

Response: Thank you for your comments. The information about DMTS was added in the discussion section.

-Table 2 lacks statistical parameters. Convert ppb in mass/mass. Include information such as the number of samples, for example.

Response: Thank you for your comments. The statistical parameter was added in the legend of Tables. Ppb means µg/kg here. The information about numbers, groups, detecting methods was added in materials and methods sections. 

Best wishes.

Reviewer 3 Report

The Authors presented an article entitled “Dimethyl Trisulfide (DMTS), a volatile compound produced 2 by Pseudomonas stutzeri YM6 and its inhibitory effect on As- 3 pergillus flavus growth and aflatoxins production in grains”.

Although the paper is interesting, I find that there are some minor issues that require addressing prior to this being considered. I have identified the main points for consideration below:

General comments

1.      Materials and methods section could be improved by mention the name of Laboratory where experiments were carried out.

2.      Line 174-176: The aflatoxins in peanut and maize samples were also determined through quantitative analysis. In the control group, the total amounts of aflatoxins in peanut samples were 175 99.491, 330.165, and 1767.619 ppb at aw of 0.740, 0.859, and 0.923, respectively……which method was used to determine the aflatoxins content in peanuts and maize?

3.      Which are the maximum admitted limits for aflatoxins in maize and peanuts? I think it would be useful to mention the maximum limits allowed in corn and peanuts intended for human and animal consumption.

Author Response

Dear reviewer:

  On behalf of my co-authors, we thank you very much for giving us an opportunity to revise our manuscript, we appreciate reviewers very much for your constructive comments and suggestions on our manuscript. The point-to-point response to reviewer 3 as follows:

  The Authors presented an article entitled “Dimethyl Trisulfide (DMTS), a volatile compound produced by Pseudomonas stutzeri YM6 and its inhibitory effect on As-  pergillus flavus growth and aflatoxins production in grains”.

Although the paper is interesting, I find that there are some minor issues that require addressing prior to this being considered. I have identified the main points for consideration below:

 General comments

  1. Materials and methods section could be improved by mention the name of Laboratory where experiments were carried out.

Response: Thank you for your comments. The sentence ‘All experiments were carried out in Molecular Biotechnology Laboratory of Triticeae Crops, Huazhong Agricultural University, Wuhan, China.’ was added in the materials and methods section.

  1. Line 174-176: The aflatoxins in peanut and maize samples were also determined through quantitative analysis. In the control group, the total amounts of aflatoxins in peanut samples were 175 99.491, 330.165, and 1767.619 ppb at aw of 0.740, 0.859, and 0.923, respectively……which method was used to determine the aflatoxins content in peanuts and maize?

Response: Thank you for your comments. The aflatoxins content was determined through ultra-performance liquid chromatography and mass spectrometry. The message was added in materials and methods section.

  1. Which are the maximum admitted limits for aflatoxins in maize and peanuts? I think it would be useful to mention the maximum limits allowed in corn and peanuts intended for human and animal consumption.

Response: Thank you for your comments. It was very helpful to us. The maximum limit in corn and peanut was added in the ‘introduction’ section.